# Effect of Preheating and Post-Heating on the Microstructures and Mechanical Properties of TC17-Ti_2_AlNb Joint with Electron Beam Welding

**DOI:** 10.3390/ma17071654

**Published:** 2024-04-03

**Authors:** Lihang Li, Pengfei Fu, Bochao Lin, Xuedong Wang

**Affiliations:** Key Laboratory of Science and Technology on Power Beam Processes, AVIC Manufacturing Technology Institute, Beijing 100024, China; fupengfei97@163.com (P.F.); 18524439219@163.com (B.L.); wxue2012@yeah.net (X.W.)

**Keywords:** electron beam welding (EBW), dissimilar materials, heat treatment, microstructure, mechanical property

## Abstract

To enhance welding quality and performance, preheating and post-heating are usually employed on high-temperature materials, concurrently with welding. This is a novel technique in vacuum chamber electron beam welding (EBW). TC17 and Ti_2_AlNb alloys are the hot topics in aero-engine parts, and the welding of dissimilar materials is also a broad prospect. To settle welding cracks of Ti_2_AlNb, EBW with preheating and post-heating was investigated on TC17 and Ti_2_AlNb dissimilar alloy, which improved the manufacturing technology on high-temperature materials. The dissimilar joint no longer had cracks after preheating, which exhibited excellent welding stability and metallurgical homogeneity, and preheating and annealing had an important effect on mechanical properties. The joint strength after 630 °C annealing is higher than that of TC17 alloy base metal (BM) and other annealing temperatures, reaching 1169 MPa at room temperature and 894 MPa at 450 °C tensile condition. The joint plasticity after 740 °C annealing is equivalent to TC17 BM. EBW with preheating improved the microstructure characteristics and enhanced the plasticity of Ti_2_AlNb alloy weld and dissimilar joint, which would contribute to the application of Ti_2_AlNb alloy and Ti_2_AlNb dissimilar parts.

## 1. Introduction

Titanium alloys are widely used in many fields, such as aerospace, chemical industry, shipbuilding and biomedical science, because of their low density; high specific strength; and comprehensive properties, such as good corrosion resistance and fatigue properties [1,2]. TC17 titanium alloy is a typical α + β two-phase titanium alloy, which is often used to manufacture key components of aircraft engines, such as shafts, compressor disks and blades, due to its high strength, good fracture toughness, high hardening depth and wide forging temperature range [3,4]. Ti_2_AlNb alloy originates from Nb-modified Ti_3_Al-based alloy, mainly consisting of B2, O and α2 phases, and it has good plasticity, fracture toughness and creep resistance, which is expected to replace nickel-based superalloy for aero-engine compressors [5,6,7]. In order to meet the future aero-engine demand for high-performance lightweight and high-temperature resistant materials, TC17 and Ti_2_AlNb’s dissimilar structure will be used as an alternative to the high-temperature part of aircraft engines. Solving the welding problem of these two alloys can not only promote the application in high thrust-to-weight ratio engines, but also be important to reduce the weight of the aircraft and improve the high-temperature service performance of the aircraft.

Due to the different chemical composition and intermetallic compound, the welding of dissimilar materials is usually prone to brittle phase, segregation and large residual stresses [8,9,10]. Vacuum EBW offers higher benefits due to its concentrated energy density, tiny welding deformation and high welding purity, as well as its process adaptability for welding dissimilar materials. Studies have shown that an excessive temperature gradient during Ti_2_AlNb alloy welding was prone to cracking and welding defects [11]. Because of the high residual stress during the heat treatment process after welding, the secondary crack will also appear further [12,13]. The mechanical properties of the welded joints between different materials are anticipated to improve thanks to the excellent weldability of TC17 titanium alloy [14,15].

Preheating is an effective method to eliminate welding cracks for Ti_2_AlNb alloy with EBW, such as multi-beam preheating and scanning preheating [16,17]. However, the multi-beam scanning preheating gradually diminishes when the weld thickness exceeds 5 mm. It is expected that the overall preheating method will perform better for larger thickness because the temperature uniformity of the material along the thickness direction is better. Furthermore, the preheating temperature needs to be matched to the post-weld annealing temperature. The preheating also has impacts on the grain and precipitation phase of the joints, which would improve the mechanical properties [17].

The effect of preheating and post-weld annealing on TC17-Ti_2_AlNb dissimilar materials with EBW was investigated in this paper. The influence of various preheating and annealing temperatures on the microstructure and mechanical properties were investigated in this study. The strengthening mechanism of precipitation phase was discussed during annealing, and the preheating and annealing procedure was optimized accordingly, which helped to enhance the performance of welded joints. The better preheating effect is achieved by installing the preheating device directly in the vacuum chamber and performing electron beam welding immediately after preheating. The addition of TC17 also improves the weldability of the joints, which ultimately effectively inhibits the generation of welding cracks. This method can be further extended to the welding of any kind of refractory crack-prone materials, which is significant in engineering applications.

## 2. Materials and Methods

The materials of TC17 titanium alloy (Ti-5Al-4Mo-4Cr-2Zr-2Sn) and Ti_2_AlNb alloy (Ti-22Al-27Nb-0.5Mo) were forged by the Western Superconducting Technologies Company and the Central Iron & Steel Research Institute from China, respectively. The plate dimension was 150 mm × 100 mm × 10 mm. The investigation of the welding process was used by ZD150-30CCV9M from China, with 30 kW power and 150 kV voltage.

The schematic of overall preheating device used in the vacuum chamber is shown in Figure 1, which was a significant part to the process for vacuum thermal protection. The device was mainly composed of a platform, heating tape and thermal insulation. The welded plate was assembled on a platform during preheating and was encircled by a heat insulation chamber to prevent heat loss. The EBW process was employed on the plate from the top after preheating. The platform was heated by heating elements below, and the thermocouples were used to monitor the temperature to guarantee the precision temperature. Before electron beam welding, the two thermocouples were to be maintained at a temperature difference of no more than 10 °C for at least 40 min. Post-weld annealing was carried out in a vacuum furnace. Different procedures were used to match the preheating and annealing temperatures. Preheating temperatures were set at 20 °C (not preheated) and 400 °C, and post-weld annealing temperatures were respectively set at 630 °C, 740 °C, and 850 °C, as shown in Table 1.

Figure 2a,b showed the typical surface characteristics of the welds. After X-ray inspection, the welding quality was conformed to meet the standard of Ⅰ class, and the joints had no crack defects with preheating. The schematic diagram of the electron beam welding process is shown in Figure 2c. The surface of the test plate was polished and cleaned before welding, and then it was clamped to the welding fixture and assembled to the welding equipment platform. The optimized welding parameters were the acceleration voltage of 120 kV, welding current of 26 mA and welding speed of 10 mm/s. 

The specimens were prepared for metallographic observation, were ground and polished, and then were corroded using newly formulated Kohler’s reagent. ZEISS field emission scanning electron microscopy (SEM) from Germany was used for high-magnification microstructure observation. The instrument used for electron backscatter diffraction (EBSD) was model JSM-7900F from Japan, and the probe was EDAX HIkari XP. The tensile tests were conducted on the CSS-2220 electronic universal testing machine from China under room temperature and high-temperature conditions. As the main service temperature of TC17 was 450 °C, the high-temperature tensile properties were chosen to be tested at 450 °C. The dimensions of the tensile specimen are shown in Figure 2d. The elongation is the percentage of the ratio of the residual elongation according to the mark (L-L_0_) after fracture to the original mark (L_0_). For the determination of elongation after fracture, the broken parts of the specimen are carefully mated together so that their axes are in the same straight line, and measures are taken to ensure that the broken parts of the specimen are in proper contact with each other before measuring the specimen’s distance after fracture (L).

The finite element analysis of the welding temperature field was completed by the SYSWELD software from ESI company, France. The heat source of Gaussian three-dimensional volume was employed in modeling the welding process of TC17-Ti_2_AlNb dissimilar materials, including the conventional welding and welding with preheating. The Volume of Fluid (VOF) method was used to calculate the welding velocity field. The three-dimensional model was established according to the plate dimension. The initial and boundary conditions were based on the following assumptions: (1) The material was continuous, homogeneous and isotropic. (2) The physical characteristics of the material depended on the temperature. (3) The heat dissipation under vacuum conditions was heat conduction between the test plates, and heat radiation only generated on the surface. There was no convective heat transfer. (4) The vaporization and flow of liquid metal during the experiment were not considered.

## 3. Results and Discussion

The morphology of the TC17-Ti_2_AlNb alloy with EBW is shown in Figure 3a,b. The weld and base metal (BM) exhibited discontinuous microstructure characteristics. The welded joint comprised a BM zone near the TC17 side (TC17-BM), HAZ near the TC17 side (TC17-HAZ), fusion zone (FZ), HAZ near the Ti_2_AlNb side (Ti_2_AlNb-HAZ) and BM zone near the Ti_2_AlNb side (Ti_2_AlNb-BM), as seen in Figure 3c–e, which show the weld cracks appearing in the heat-affected zone on the Ti_2_AlNb side after annealing without preheating. The locations where cracks were found are marked with a yellow box in Figure 3a. This crack appeared during the annealing process, sprouting from the Ti_2_AlNb-HAZ and expanding along the intergranularity to both sides. The direction of the crack extension is basically perpendicular to the weld depth direction, indicating that the crack is generated due to the large temperature gradient between the FZ and Ti_2_AlNb-BM.

The microstructure of the TC17 alloy shows an equiaxial α phase distributed on the β phase (Figure 4a). The microstructure of the Ti_2_AlNb alloy is characterized by an α phase and O phase distributed on the β-phase base (Figure 4b). The dissimilar joint microstructures of EBW (as-welded) and EBW with heat treatment are shown in Figure 4 and Figure 5. The microstructure’s evolution is mainly attributed to the rapid melting and solidification process during EBW, but the preheating prior to welding had little impact on the microstructure. The weld (Figure 4c) retained a significant amount of β phase due to the fast cooling after welding. Figure 4d shows the microstructure of the FZ after annealing. O or α phases precipitated from the β phase were uniformly distributed in the weld, with the majority of the O phase precipitating from the α phase’s grain boundaries and creating a marginal O phase around the α phase [18,19].

A small amount of α phase existed in HAZ of the dissimilar joint (Figure 5a,b). The microstructure displayed a more pronounced variability when the precipitation phase gradually appeared after annealing. The microstructures of TC17-HAZ (Figure 5c) were a lot of precipitated strips of the α phase. The reticulated B2 + α2 + O microstructures existed in Ti_2_AINb-HAZ (Figure 5d) with a high proportion of O phase.

In contrast to conventional welding, preheating increases the temperature and improves the homogeneity of the joints’ temperature field, which had a significant impact on the welding thermal cycling. The HAZ exhibited a longer post-weld cooling time and a slower cooling rate, which was verified from the modeling in Figure 6. Figure 6a,b, respectively, show the simulation results of the temperature field under two conditions: non-preheated and preheated. From Figure 6c, the peak temperature of the location with 3 mm distance from the welding center increased at least 200 °C in the preheating area during welding. The preheated welding took 19.5 s as opposed to 4.2 s for the conventional welding under the same conditions, cooling to 600 °C. To better simulate the changes in keyhole and melting pool during the EBW process, the Volume of Fluid (VOF) method was used to calculate the welding velocity field [20]. The model was optimized and validated by comparing the shapes in the model with the cross-section of the welded joint. It could be found from Figure 6d,e and Table 2 that the shapes of the transverse cross-sections of the weld obtained were in accordance with the experimental results for both conditions, and the calculation errors were restricted to 7.8%.

The results of the grain size characterized by EBSD also supported the results of the simulation (Figure 7). The grain sizes in each region increased significantly after welding compared to the base metal. The average grain sizes of TC17-BM and Ti_2_AlNb-BM are about 15 μm and 2 μm, respectively (Figure 7a). In general, the grain size basically shows a decreasing tendency with the increase in the annealing temperature, and the change is especially obvious in the Ti_2_AlNb-HAZ. However, under the condition of non-preheated, the FZ and TC17-HAZ of the welded joints were characterized by small fluctuating changes.

Comparing the grain size under NP-welded and P-welded conditions, the grains were more uniform after preheating (Figure 7b). Preheating lowered the gradient of temperature throughout the welding zone, which slowed post-weld cooling and lengthened the time for grain nucleation and growth. The grains for the P-welded conditions were coarser than those of the NP-welded conditions, which were more uniform in the HAZ and FZ. As the post-weld annealing process moved on, the region’s average grain size tended to decrease while simultaneously becoming more uniform.

The tensile properties of TC17-Ti_2_AlNb joints at room temperature are shown in Figure 8a,b. The tensile strength was attained for 1169 MPa under the P-630 condition, which was roughly equivalent to the strength of the Ti_2_AlNb base metal. The preheated joint’s tensile strength was slightly higher at 850 °C than at 740 °C, but the joint’s elongation displayed the opposite tendency. The preheating method effectively improved the tensile strength of the joints at room temperature by 5%, 5.2% and 3.3% at 630 °C, 740 °C and 850 °C, respectively. The tensile fracture at 740 °C annealing was in the TC17-HAZ, while another two temperatures’ fractures were both in the FZ, as shown in Figure 9a,b.

Figure 8c,d demonstrate the tensile properties at 450 °C of the TC17-Ti_2_AlNb joints. It could be found that the strength of TC17-Ti_2_AlNb joints at 450 °C varies with different annealing temperatures in the same way as the room temperature, showing a slightly higher strength at 630 °C and the lowest strength at 740 °C. The tensile strength reached 894 MPa under the P-630 condition. Compared with non-preheated process, the tensile strength of the dissimilar joints increased by 4%, 5.3% and 4% at 630 °C, 740 °C and 850 °C, respectively. The elongation reached over 18% at 740 °C and 850 °C, which is comparable to the Ti_2_AlNb base metal. The elongation is 15.8% for NP-630 and 15.5% for P-630. The fractures at 450 °C are all located in the TC17-BM, as shown in Figure 9c. The preheating proved to improve the high-temperature mechanical properties of the dissimilar joints effectively.

The increased mechanical properties of the dissimilar joints were attributed to the improvement of microstructure and grain structure. Figure 10a shows, for the condition of the 850 °C treatment, the proportion of phase composition in each area of non-preheated welding and preheated welding characterized by EBSD. The most noticeable change was that the preheating reduced the proportion of the α phase in the FZ from 45.4% to 16.6%. Figure 10b–d show the phase ratio of the TC17-Ti_2_AlNb joint for three annealing temperatures, respectively. Preheated and non-preheated conditions are both presented as a large proportion of the β phase after welding. However, the ratio of O phase that precipitated from the β-phase matrix would differ after various annealing temperatures, which would affect the mechanical properties of the dissimilar joints.

The tensile properties of dissimilar joints with different annealing temperatures are closely related to the microstructures and phase ratios. The FZ maintained β-phase characteristics after 630 °C annealing, showing higher strength but lower plasticity at room temperature, and the fracture location was also in the FZ zone. The precipitated phases first appeared in FZ after 740 °C annealing. Both the α phases and O phases were tiny, and the β phase remained dominant in the zone. However, after 850 °C annealing, the O and α phases in the FZ zone reached 7.7% and 16.6%, respectively. Generally, Ti_2_AlNb-based alloys with single O phases had lower strength and plasticity due to the large proportion of O/O grain boundaries [21]. The α/B2 grain boundary and α/O grain boundary, however, were more easily to develop cracks when the α phase emerged as a brittle phase in FZ, which ultimately caused the joint to fracture with the crack growth [22,23].

The HAZ on both sides grew a large proportion of precipitated phases during annealing. The precipitation of α phase in the TC17-HAZ zone showed a trend of increasing and then decreasing, with the highest proportion of α phase after 740 °C annealing and reaching 75%. This resulted in the welded joints’ rupturing on the TC17 side, with a low room temperature tensile strength but high elongation. The reason could be attributed to the sub-stable β phase decomposing and secondary needle-like α phase precipitating after 630 °C and 740 °C annealing [23,24]. However, when the annealing temperature rose to 850 °C, which was located at the higher temperature of the α + β phase region, the annealing process resulted in an α→β phase transformation, converting the majority of the α phase into the β phase.

In Ti_2_AlNb-HAZ, the annealing temperature of 630~850 °C was also in the B2 + O two-phase region from the Ti_2_AlNb phase diagram [25]. The driving force of O-phase nucleation and growth in the joint increased with the temperature enhancing, and the resulting O phase would be spread around the α2 phase, inhibiting the α2 phase growth further [23,26]. The superior performance of Ti_2_AlNb under a high temperature resulted in a welded joint with a more excellent high-temperature strength than the TC17 base metal, eventually causing the fractures in TC17-BM [27]. The increase in the proportion of the α phase in the TC17 alloy is the main reason for the increase in plasticity. Moreover, 630 °C just reaches the α + β phase transforming temperature of the TC17 alloy, at which less α phase is generated after heat treatment. The effect of preheating on the α phase is less pronounced compared to higher temperatures (740 °C and 850 °C), so that the high-temperature plasticity remains essentially unchanged.

Figure 11a,b show the tensile fracture at room temperature when the annealing temperature is 630 °C and 850 °C. The morphology showed typical microporous intensive fracture characteristics, and the fracture was located in FZ. Lacerate ribs consisting of dimples can be observed at the fracture, indicating that the welded joints have a certain plastic deformation capacity at room temperature, and, eventually, a ductile fracture was caused by local plastic deformation. The tensile fracture location at 740 °C is located on the TC17 side, and an obvious necking phenomenon occurs in Figure 11c,d. The proportion of the α phase on the TC17 side after annealing at this temperature increased significantly, reaching 74% (Figure 10d), so that the deformation in the tensile tests was more distributed to the hexagonal structure, which was more favorable to the opening of multiple slip systems, and eventually manifested as a significant increase in the plasticity of the joint. However, the strength of the α phase is not as strong as that of the β phase, so the tensile strength of the joint is significantly reduced at 740 °C, as seen in Figure 8a. The high-temperature tensile fractures in Figure 11e,f were all located on the TC17 side, and the morphology was similar to that of Figure 11a,b. The reason for this phenomenon was mainly due to the precipitation of secondary α phase during the high-temperature test so that the α phase dominated the microstructure of the TC17 side. 

Due to the longer post-weld cooling time caused by preheating, the proportion of annealing precipitation phases in the joints was significantly reduced. Figure 12 showed the grain orientation polar chart of TC17-Ti_2_AlNb joints with non-preheated and preheated welding processes. The grain orientation distribution of the joint without preheating was more stochastic, indicating that the growth of grains was not obvious in terms of directionality. The greatest texture strength was 15.622 for preheated welding, with grain growth being more concentrated in the (001) direction. Compared with the non-preheated welding, the significant increase in texture strength indicated that the welding microstructure exhibited significant directionality under preheated welding. The change in grain orientation could also be attributed to the grain remaining in the high temperature zone for a long time. As a result, the cooling rate was slower, and the grain was more likely to be selective in growth.

In general, the microstructure of FZ usually emerged as the atomically disordered β phase after solidification, and it was then further ordered by a solid-state phase transformation during subsequent cooling [17,28]. In contrast to preheated conditions, the rapid cooling rate without preheating prevented the orderly transformation of β phase, leaving a more chaotic sub-stable β phase. The O and α phases precipitated after annealing mainly originated from the decomposition of sub-stable β phase. The transforming of B2→O for Ti_2_AlNb side and β→α for TC17 side would arise in FZ when the annealing temperature reached 600 °C. Therefore, the slow cooling rate and less residual sub-stable β phase of EBW with preheating would lead to a decrease in the proportion of O phase and α phase after annealing. The reduction of the α phase was the main reason for the increase in the joint’s strength under preheating conditions.

The tensile strength and high-temperature plasticity of the TC17-Ti_2_AlNb joint could be effectively improved by appropriate combination of the preheating and post-weld annealing. As the TC17 titanium alloy with greater weldability was added, we could raise the joint’s tensile strength and improve its mechanical properties in comparison to Ti_2_AlNb electron beam-welded joints individually [19].

## 4. Conclusions

(1)EBW with preheating was employed TC17 and Ti_2_AlNb dissimilar alloy, which showed excellent welding metallurgical compatibility with non-defects joint after preheating. The weld has a mainly β-phase microstructure, and a few of O phases and α phases were generated in the weld only after annealing. But a lot of O and α phases precipitated in HAZ on TC17 and Ti_2_AlNb sides.(2)The tensile strengths of dissimilar joints under room temperature and 450 °C were equivalent to those of TC17 BM, and the strengths after 740 °C annealing are lower than those after 630 °C and 850 °C annealing. The tensile fracture after 740 °C annealing was in TC17 HAZ due to the low strength, which reaches 75% proportion of α phase in HAZ. The dissimilar joint achieved the highest strength after 630 °C annealing, reaching 1169 MPa at room temperature and 894 MPa at 450 °C tensile condition.(3)The preheating increased the peak temperature in HAZ and prolonged the welding cooling period from the numerical simulation, which decreased almost 30% of brittle α phase, contributing to an increase of 5% in tensile strength. Preheating would effectively increase the strength and plasticity of the dissimilar joint.

## Figures and Tables

**Figure 1 materials-17-01654-f001:**
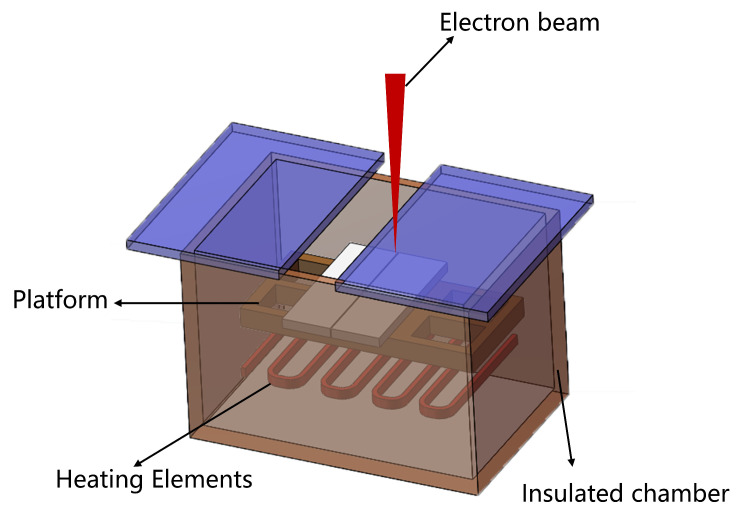
The schematic of preheating device in vacuum electron beam welding machine.

**Figure 2 materials-17-01654-f002:**
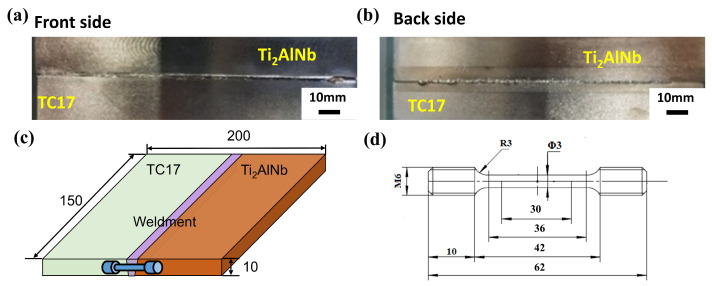
(**a**) Front side and (**b**) bake side of TC17-Ti_2_AlNb dissimilar welds. (**c**) Schematic diagram of electron beam welding process. (**d**) Size of tensile test specimen at room temperature and 450 °C.

**Figure 3 materials-17-01654-f003:**
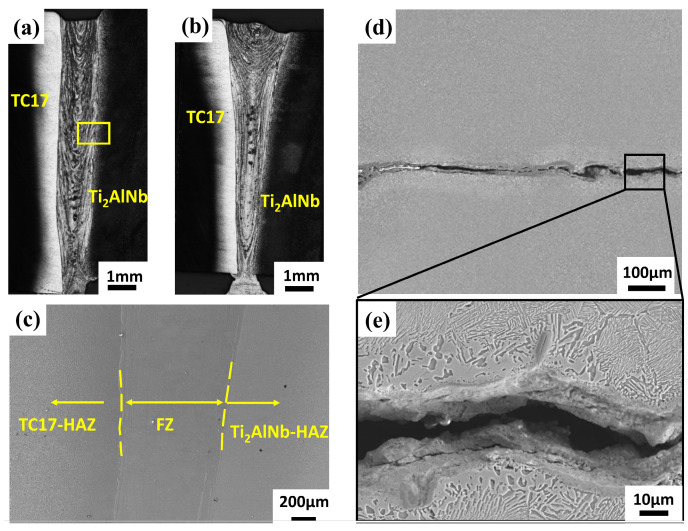
Cross-section morphology of TC17-Ti_2_AlNb welded joint: (**a**) macro view of non-preheated joint and (**b**) preheated joint by optical microscope. (**c**) Micro view by SEM. The cracks of TC17-Ti_2_AlNb welded joint on Ti_2_AlNb side after annealing without being preheated by SEM: (**d**) macro view and (**e**) micro view.

**Figure 4 materials-17-01654-f004:**
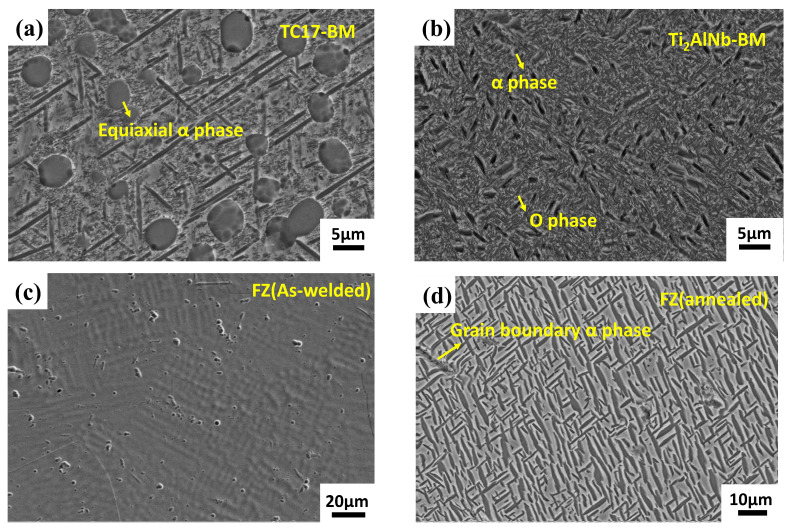
Microstructure of base metal by SEM: (**a**) TC17 alloy and (**b**) Ti_2_AlNb alloy. Microstructure of TC17-Ti_2_AlNb electron beam-welded joints for FZ by SEM: (**c**) as-welded and (**d**) annealed.

**Figure 5 materials-17-01654-f005:**
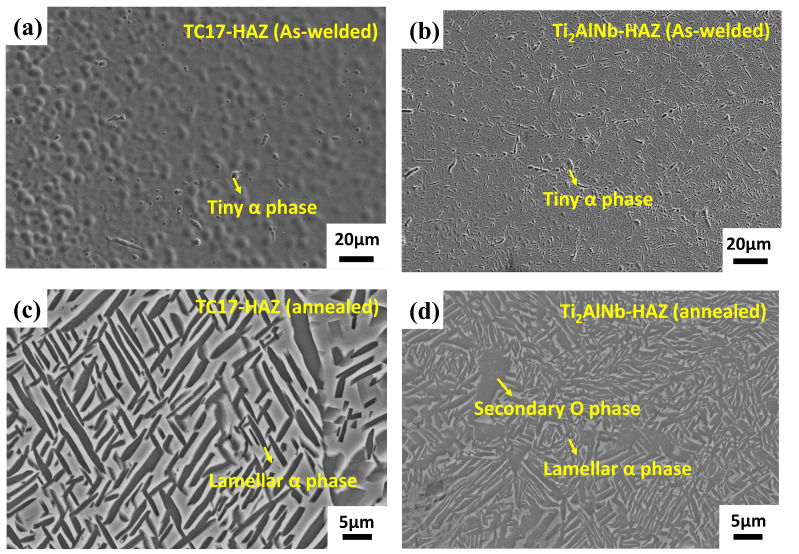
Microstructure of TC17-Ti_2_AlNb electron beam-welded joints for HAZ by SEM: (**a**) TC17-HAZ (as-welded), (**b**) Ti_2_AlNb-HAZ (As-welded), (**c**) TC17-HAZ (annealed) and (**d**) Ti_2_AlNb-HAZ (annealed).

**Figure 6 materials-17-01654-f006:**
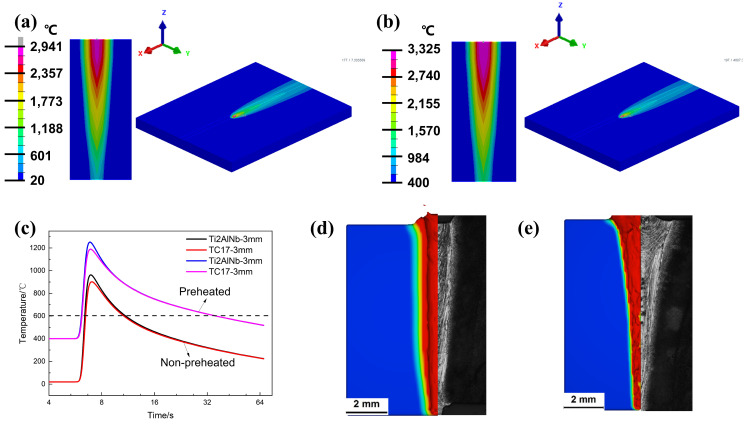
The temperature field of (**a**) non-preheated and (**b**) preheated. (**c**) Temperature field variation at 3 mm nodes from the center of the weldment on both sides after welding. The calculational and experimental results of the transverse cross-sections of the weld: (**d**) non-preheated and (**e**) preheated.

**Figure 7 materials-17-01654-f007:**
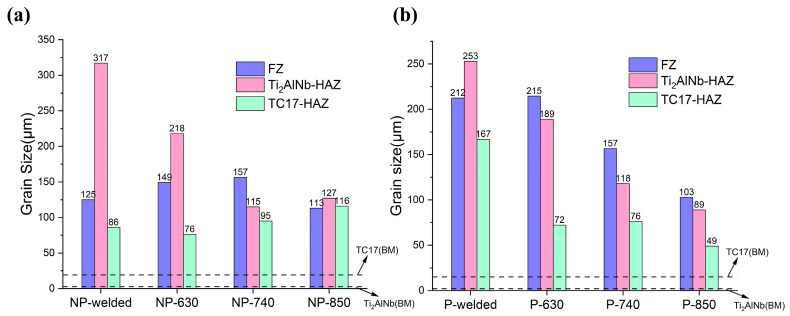
Grain size distribution of TC17-Ti_2_AlNb dissimilar joints: (**a**) non-preheated and (**b**) preheated.

**Figure 8 materials-17-01654-f008:**
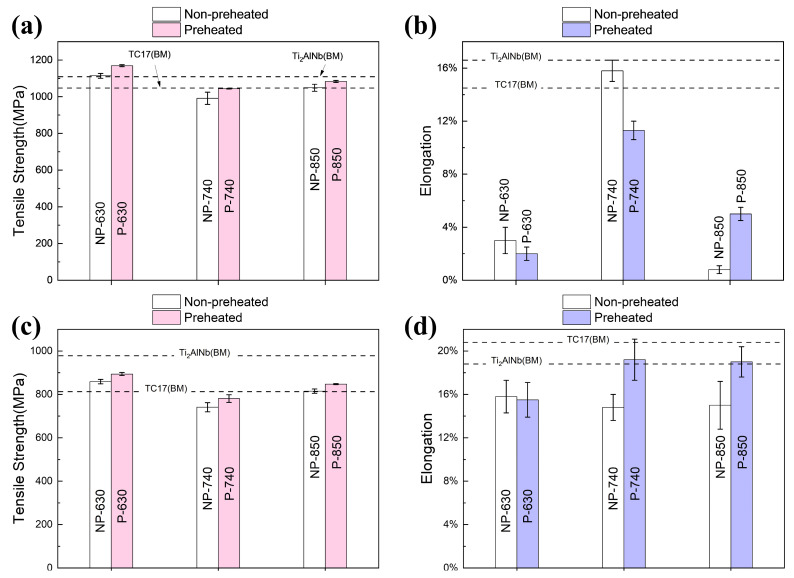
Tensile properties of TC17-Ti_2_AlNb joints at room temperature: (**a**) tensile strength. (**b**) Elongation and high temperature tensile properties of TC17-Ti_2_AlNb joints at 450 °C: (**c**) tensile strength and (**d**) elongation.

**Figure 9 materials-17-01654-f009:**
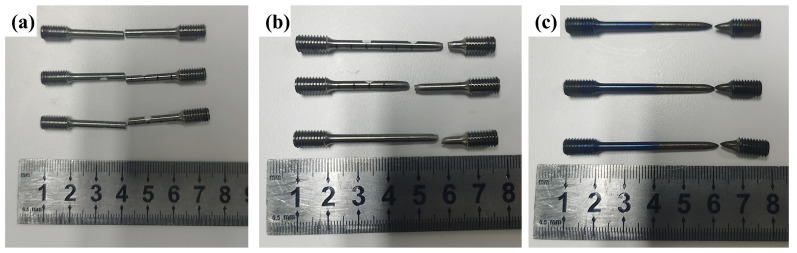
Tensile specimen of TC17-Ti_2_AlNb joints: (**a**) room temperature for 630 °C × 2 h AC and 850 °C × 2 h AC, (**b**) room temperature for 740 °C × 2 h AC and (**c**) high temperature at 450 °C.

**Figure 10 materials-17-01654-f010:**
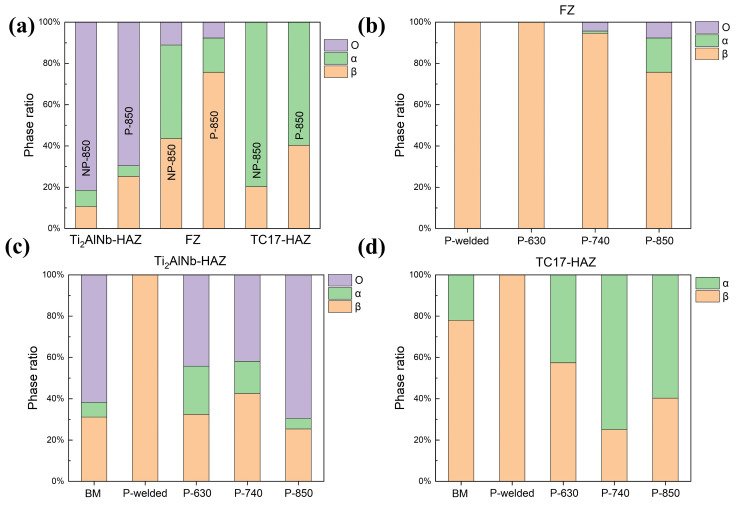
Histograms of phase ratios in TC17-Ti_2_AlNb joints in different conditions. (**a**) Comparison of preheated and non-preheated in different regions of the weld. Comparison of different conditions for (**b**) FZ, (**c**) Ti_2_AlNb-HAZ and (**d**) TC17-HAZ.

**Figure 11 materials-17-01654-f011:**
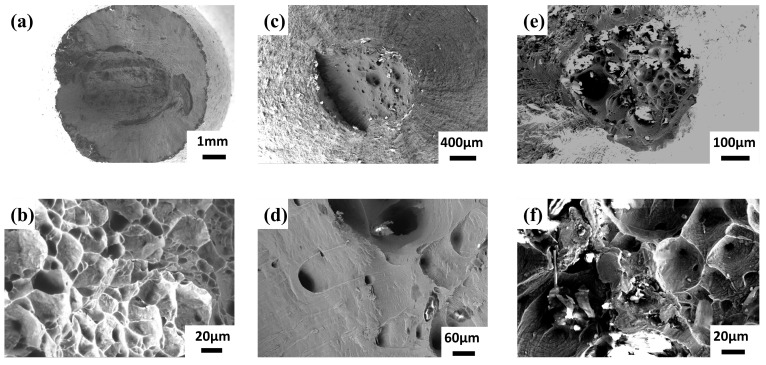
Tensile fracture of TC17-Ti_2_AlNb joints by SEM: (**a**,**b**) room temperature for 630 °C × 2 h AC and 850 °C × 2 h AC, (**c**,**d**) room temperature for 740 °C × 2 h AC and (**e**,**f**) high temperature at 450 °C.

**Figure 12 materials-17-01654-f012:**
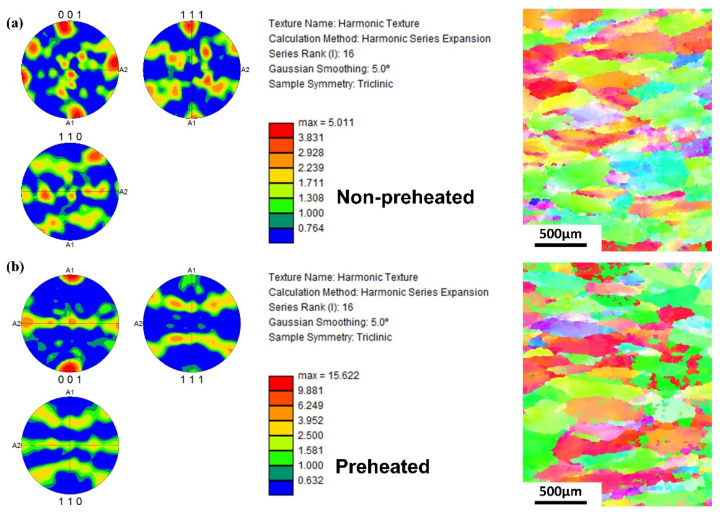
EBSD orientation polar map and grain structure of TC17-Ti_2_AlNb welded joints for (**a**) non-preheated and (**b**) preheated.

**Table 1 materials-17-01654-t001:** The preheating and annealing process of TC17-Ti_2_AlNb EBW joint. (VAC, vacuum condition; and AC, air cooling).

Number	Preheating Process	Annealing Process
NP-welded	20 °C, VAC	/
NP-630		630 °C × 2 h, AC
NP-740		740 °C × 2 h, AC
NP-850		850 °C × 2 h, AC
P-welded	400 °C, VAC	/
P-630		630 °C × 2 h, AC
P-740		740 °C × 2 h, AC
P-850		850 °C × 2 h, AC

**Table 2 materials-17-01654-t002:** The comparison of the calculational and experimental results of the weld dimensions.

Location	Non-Preheated (Figure 6d)			Preheated (Figure 6e)		
	Experimental Result/mm	Calculational Result/mm	Error	Experimental Result/mm	Calculational Result/mm	Error
Width of the top of the weld	1.26	1.21	4.3%	3.23	2.99	7.8%
Width of the intermediate part of the weld	0.98	1.03	5%	1.26	1.25	1.0%
Width of the bottom of the weld	0.59	0.62	4.3%	0.62	0.59	4.3%

## Data Availability

Data are contained within the article.

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
