# Peer review of "Effect of Preheating and Post-Heating on the Microstructures and Mechanical Properties of TC17-Ti2AlNb Joint with Electron Beam Welding"

_materials, 2024, doi:10.3390/ma17071654_

Round 1

Reviewer 1 Report

Comments and Suggestions for Authors

All recommendations are in the attached file.

Reviewer 2 Report

Comments and Suggestions for Authors

Very interesting article. The article examined welded joints made without and with pre-heating and subjected to an annealing process at specific temperatures.

The research methodology attempts to describe the heating and annealing conditions, but it is not understandable. Due to the many factors analyzed in the work, I propose to use a diagram that will be more understandable to the reader. At the same time, please include the heating and cooling conditions after processing in the diagram.

A dissimilar joint based on two grades of titanium was tested. It is worth including microstructures for each of them, which will present the starting material after welding and heat treatment. The analysis of the results indicates the changes that occur during the annealing process, while there is no reference point. Please complete these results for each state.

Please indicate the places where cracks occur after welding on the macrostructure.

A transverse tensile test was performed and parameters were determined for each material. Please indicate how it was done if the fracture always occurs only in one material (the weakest). What more, it is presented in conclusions. Additionally, please provide information about the location of the break and the nature of the break. The high temperature of the experiment and the presence of cracks after welding should be visible in the fractures - please post photos of the fractures.

How was the extension (elongation) determined?

The analysis used a Gaussian model for the heat source, which distributes heat evenly, while in EBW welding the distribution is uneven and the center is heated much more strongly (key-hole is created to weld 10 mm plate) than adjacent areas. This is due to the way the electron beam affects the material. Have you considered using the Goldak or hybrid model to describe the heat source?

Please pay attention to the font in all photos - the scales are illegible and require improvement - this applies to all drawings.

Please explain the grain size in HAZ of 317 um. How does grain refinement occur at a temperature of 850 C? Please complete the results for the remaining annealing temperatures.

Round 2

Reviewer 1 Report

Comments and Suggestions for Authors

Good research work. ¡Congratulations!

Reviewer 2 Report

Comments and Suggestions for Authors

Thank you for your answers and improvments of paper.